# The Key Targets of NO-Mediated Post-Translation Modification (PTM) Highlighting the Dynamic Metabolism of ROS and RNS in Peroxisomes

**DOI:** 10.3390/ijms25168873

**Published:** 2024-08-15

**Authors:** Ulugbek Ergashev, Mei Yu, Long Luo, Jie Tang, Yi Han

**Affiliations:** 1National Engineering Laboratory of Crop Stress Resistance Breeding, School of Life Sciences, Anhui Agricultural University, Hefei 230036, China; ergashevulugbek207@gmail.com (U.E.); yumei@ahau.edu.cn (M.Y.); luoloong@ahau.edu.cn (L.L.); 2Anhui Provincial Key Laboratory of Hazardous Factors and Risk Control of Agri-Food Quality Safety, School of Resources and Environment, Anhui Agricultural University, Hefei 230036, China

**Keywords:** nitric oxide (NO), reactive nitrogen species (RNS), *S*-nitrosoglutathione (GSNO), peroxynitrite (ONOO^−^), post-translational modification (PTM), H_2_O_2_

## Abstract

Nitric oxide (NO) has been firmly established as a key signaling molecule in plants, playing a significant role in regulating growth, development and stress responses. Given the imperative of sustainable agriculture and the urgent need to meet the escalating global demand for food, it is imperative to safeguard crop plants from the effects of climate fluctuations. Plants respond to environmental challenges by producing redox molecules, including reactive oxygen species (ROS) and reactive nitrogen species (RNS), which regulate cellular, physiological, and molecular processes. Nitric oxide (NO) plays a crucial role in plant stress tolerance, acting as a signaling molecule or free radical. NO is involved in various developmental processes in plants through diverse mechanisms. Exogenous NO supplementation can alleviate the toxicity of abiotic stresses and enhance plant resistance. In this review we summarize the studies regarding the production of NO in peroxisomes, and how its molecule and its derived products, (ONOO^−^) and *S*-nitrosoglutathione (GSNO) affect ROS metabolism in peroxisomes. Peroxisomal antioxidant enzymes including catalase (CAT), are key targets of NO-mediated post-translational modification (PTM) highlighting the dynamic metabolism of ROS and RNS in peroxisomes.

## 1. Introduction

Plants as stationary organisms, encounter various difficult environmental circumstances during their growth stages, such as abiotic stressors like drought, salinity, extreme temperatures, and heavy metals, among others, as well as biotic stresses such as bacterial, fungal, viral infections, and herbivory. In response to these challenges, plants activate a range of molecular, cellular, and physiological adaptations [1]. Thus, at the cellular level, plant reactions to their surroundings are highly intricate, entailing interactions and communication across numerous molecular pathways. Among the initial reactions, plants employ reactive oxygen species (ROS) and reactive nitrogen species (RNS) as vital signaling molecules, orchestrating various plant functions by triggering secondary messenger activation, inducing gene transcription, and modifying enzyme activity [2]. Oxidative stress leads to a disruption in redox balance due to the overproduction of ROS and RNS. While both ROS and RNS play crucial roles in plant stress responses, information on RNS generation under oxidative stress, their interactions with cellular components, and plant nitrosative responses remain limited. Consequently, the excessive accumulation of unwanted ions and the generation of ROS/RNS exacerbate stress levels in plant cells, leading to detrimental effects such as lipid peroxidation, chlorophyll pigment loss, decreased biomass, and ultimately plant death. This results in an overall decline in crop growth and development, leading to reduced yields [3]. Under normal physiological conditions, ROS are produced through various pathways, including purine breakdown, fatty acid β-oxidation, and photorespiration [4]. Recent experimental evidence reveals that plant peroxisomes are capable of producing not just nitric oxide (NO), but also a range of related reactive nitrogen species (RNS). This group of RNS includes molecules such as peroxynitrite (ONOO^−^) and *S*-nitrosoglutathione (GSNO) [5].

Peroxisomes generate these RNS through enzymatic activities and interactions with other reactive species like superoxide anions. NO, for instance, modulates various signaling pathways and stress responses, while ONOO^−^ acts as both a signaling molecule and a potential source of oxidative damage. GSNO, a key reservoir of NO, participates in *S*-nitrosylation, influencing protein function and stress adaptation. The balance and dynamics of RNS in peroxisomes are vital for maintaining cellular redox homeostasis and coordinating responses to environmental challenges, highlighting the need for further research into their production, regulation, and functional impacts.

This review will discuss the close connections between ROS and RNS metabolism within peroxisomes. NO plays a regulatory role in this interaction by modulating the activity of certain enzymes through PTMs, particularly *S*-nitrosation (or *S*-nitrosylation) and tyrosine nitration.

The ability of peroxisomes to generate NO could have significant implications for plant cellular metabolism under normal conditions, including processes like leaf aging, pollen tube growth, and the initiation of root development induced by auxin [6]. Nevertheless, the metabolism of NO in peroxisomes becomes especially heightened during oxidative stress triggered by various abiotic factors.

## 2. Production of Oxygen Radicals in Peroxisomes

ROS represent a comprehensive grouping encompassing primarily superoxide radicals (O_2_^−^), hydroxyl radicals (OH^•^), hydroperoxyl radicals (HO_2_), alkoxyl radicals (RO) and peroxyl radicals (ROO), and additionally encompassing certain non-radical derivatives of oxygen, such as hydrogen peroxide (H_2_O_2_), singlet oxygen (1O_2_), or ozone (O_3_), alongside hypochlorous acid (HOCl) and peroxynitrite (ONOO) [7]. Certain ROS, notably hydroxyl radicals, exhibit pronounced oxidizing capabilities, swiftly targeting biological membranes and diverse biomolecules, encompassing DNA and proteins, resulting in irreversible metabolic disturbances and cellular demise [6,7]. In plants, both enzymatic and nonenzymatic antioxidative defense mechanisms are deployed across various cellular compartments. Examples of antioxidative enzymes include superoxide dismutase (SODs), catalases, peroxidases, and enzymes involved in the ascorbate-glutathione cycle. Nonenzymatic antioxidants, such as ascorbate (vitamin C), glutathione (GSH), α-tocopherol (vitamin E), β-carotene, and flavonoids, are predominantly localized in chloroplasts but are also found in other cellular organelles like mitochondria and peroxisomes. Under normal circumstances, these antioxidative defense systems in plants provide sufficient protection against ROS. However, when ROS generation surpasses the capacity of cellular antioxidant mechanisms, oxidative stress ensues. In plant cells, ROS production is primarily observed in chloroplasts, mitochondria, the plasma membrane, and the apoplastic space [8]. Biochemical and electron spin resonance spectroscopy (ESR) analyses conducted on peroxisomes extracted from pea leaves and watermelon cotyledons revealed the presence of at least two sites responsible for the generation of superoxide radicals. One site is located within the organelle matrix, where xanthine oxidase (XOD) was identified as the generating system, while another site is situated in the peroxisomal membranes and is dependent on NAD(P)H [9,10]. Xanthine oxidase aids in the oxidative transformation of xanthine and hypoxanthine into uric acid and is known as a significant producer of superoxide radicals [11]. High-performance liquid chromatography (HPLC) analysis detected xanthine, uric acid, and allantoin, the latter being a product of urate oxidase activity, in leaf peroxisomes [12]. These findings indicate that peroxisomes have a role in the cellular breakdown of xanthine, which is produced during the degradation of nucleotides, RNA and DNA [11,12]. Another site within the peroxisomal membrane responsible for generating superoxide radicals (O_2_^•−^) has been identified, potentially involving a limited electron transport chain. This chain includes a flavoprotein NADH: ferricyanide reductase with an approximate molecular weight of 32 kDa and a cytochrome b (Cyt b) [13]. The integral polypeptides of the peroxisomal membrane (PMPs) in pea leaf peroxisomes were identified through SDS-PAGE. Three of these membrane polypeptides were characterized, with molecular masses of 18, 29, and 32 kDa. These specific polypeptides were found to be responsible for generating superoxide radicals (O_2_^•−^) [14]. The primary source of superoxide radicals within the peroxisomal membrane was attributed to the 18 kDa PMP, which was suggested to potentially function as a cytochrome, likely belonging to the b-type category [15]. The 18 kDa and 32 kDa peroxisomal membrane polypeptides (PMPs) utilize NADH as the electron donor for the generation of superoxide radicals (O_2_^•−^). In contrast, the 29 kDa PMP exhibited a clear dependency on NADPH, demonstrating its capa bility to reduce cytochrome c when NADPH serves as the electron donor [15]. Based on its biochemical and immunochemical characteristics, it is highly likely that PMP32 corresponds to monodehydroascorbate reductase (MDHAR). Previous studies have detected the activity of MDHAR in the peroxisomal membranes of pea leaves [16]. The third polypeptide responsible for generating superoxide radicals, PMP29, exhibits strict dependence on NADPH called as an electron donor. It is hypothesized that PMP29 may be associated with peroxisomal NADPH: cytochrome P-450 reductase (Figure 1).

## 3. Peroxisomes Serve as Generators of Not Only ROS but Also NOS, Which Function as Signaling Molecules

The presence of NOS-like activity within peroxisomes suggests that these organelles contribute to cellular NO production. When considering this, alongside peroxisomes’ capacity for generating superoxide radicals and containing various antioxidants, a model has been proposed indicating their ability to release multiple signaling molecules like H_2_O_2_, O_2_, and NO into the cytosol. Nitric oxide, synthesized enzymatically by NOS-like activity, can interact with O_2_ radicals produced by xanthine oxidase (XOD) within the peroxisomal matrix, resulting in the formation of the potent oxidant ONOO^−^. Identification of peroxisomal proteins undergoing PTMs mediated by these NO-derived species is strong evidence of an active RNS metabolism in peroxisomes. In vivo images of NO and ONOO^−^ in Arabidopsis guard cell peroxisomes detected by confocal laser scanning microscopy (CLSM) and specific fluorescent probes. ONOO^−^, a strong oxidant and nitrating molecule involved in protein tyrosine nitration (NO_2_-Tyr), modifies protein function, mostly through inhibition [17]. This NO-derived PTM involves the covalent oxidative addition of a nitro group (-NO_2_) to tyrosine residues, a highly selective process that depends on factors such as the protein environment of the Tyr and the nitration mechanism. Interestingly, some of the proteins affected are directly involved in ROS metabolism, highlighting a close metabolic interconnection between both families of reactive species. While an increase in Tyr nitration, typically associated with nitro-oxidative stress, is an irreversible process, protein *S*-nitrosation represents a reversible regulatory mechanism occurring under both physiological and stress conditions. Peroxisomal proteins targeted by *S*-nitrosation, along with those involved in ROS metabolism, are subject to these NO-mediated PTMs. This includes proteins like the peroxisomal LON2 protease, essential for matrix protein import into peroxisomes [18]; isocitrate lyase (ICL), involved in the glyoxylate cycle; and the multifunctional AIM1-like isoform, involved in fatty acid β-oxidation. The antioxidant glutathione (GSH), a tripeptide (γ-Glu-Cys- Gly), undergoes *S*-nitrosation in order to generate GSNO, a low-molecular-weight NO reservoir, through a covalent addition of NO to the thiol group of Cys residues in order to form *S*-nitrosothiol (SNO) [19] GSNO is a key molecule given its dynamic interaction with free cysteines, GSH and proteins through processes such as *S*-nitrosation, *S*-transnitrosation and *S*-glutathionylation. Moreover, GSNO may also arise from the interaction between reduced glutathione and peroxynitrite.

However, the source of enzymatic NO, as yet unelucidated, is currently the most controversial aspect of NO metabolism in higher plants [20]. Though as yet unidentified, this protein is called NOS-like activity, as peroxisomal NO generation requires NOS proteins similar to those found in animals, including _L_-arginine, NADPH, FMN, FAD, tetrahydrobiopterin, calcium, and calmodulin [21]. The protein responsible for NO generation is imported by a type 2 peroxisomal targeting signal involving a process dependent on calmodulin and calcium [22,23] Figure 2. We tried to elucidate these origin processes with the *S*-nitrosylation and tyrosine nitration processes.

## 4. Nitric Oxide-Induced *S*-Nitrosylation, Tyrosine Nitration, Transnitrosylation, and Denitrosylation Processes

NO-mediated post-translational modifications, particularly *S*-nitrosylation and its regulation by GSH and GSNOR, are critical for maintaining redox balance, modulating signaling pathways, and enhancing stress tolerance in both animal and plant systems [24]. These mechanisms highlight the complex interplay between nitric oxide, oxidative stress, and cellular adaptation to environmental challenges. Post-translational alteration by NO occurs through one of three modification mechanisms: *S*-nitrosylation (the attachment of a nitrosothiol group to cysteine residues on target proteins) which holds pivotal roles in numerous physiological and pathological processes by modulating protein functionalities. This modification is highly conserved and extensively investigated in plants [25]. This dynamic and reversible process encompasses nitrosylation, transnitrosylation, and denitrosylation [26]. *S*-nitrosylation, also known as *S*-nitrosation in biochemistry, is regarded as a non-enzymatic mechanism whereby NO facilitates the generation of NO directly or indirectly through higher nitrogen oxides (NOx), metal-NO intermediates, *S*-nitroso compounds (SNOs), or peroxynitrite (ONOO^–^) [27]. The *S*-nitrosylation of cysteine residues within the tripeptide glutathione (GSH) results in the production of small molecular weight *S*-nitrosothiols [28]. These SNOs can subsequently serve as nitric oxide (NO) donors, contingent upon their redox potential, within physiological conditions [29]. The transfer of the NO group from one thiol to another, known as transnitrosylation, has been documented as a significant enzymatic process [30]. In this mechanism, the protein acting as the donor and facilitating the transfer of the NO group to its target is referred to as a transnitrosylase [31]. A critical factor influencing NO transfer is the disparity in redox potentials between the cysteine residues of the proteins involved, namely the donor protein-SNO and the target protein with an unoccupied thiol. The transnitrosylation process entails the reversal of the initial regulation mediated by SNO on the donor protein and can alternatively be referred to as denitrosylation [26,32]. The enzymatic and non-enzymatic facilitation of denitrosylation on target proteins plays a pivotal role in tightly regulating cysteine modification [32]. Transnitrosylation involves the activity of proteins, termed transnitrosylases, which transport and convey the NO group to its designated target [27,33].

In addition, it has been established that active multiple sclerosis (MS) patients exhibit increased NO metabolites and low molecular mass thiols in cerebrospinal fluid [34]. Additionally, there is abundant accumulation of *S*-nitrosothiols, particularly in white matter, coupled with decreased levels of glutathione (GSH). Glutathione has been identified as crucial for denitrosylation in studies using chase experiments with rat spinal cord slices, relevant to MS pathogenesis. Incubation of spinal cord slices with *S*-nitrosoglutathione demonstrated rapid disappearance of SNOs, further accelerated by increased GSH levels through GSH Ethyl Ester (GSH-EE) analogs, indicating enhanced protein denitrosylation [35]. Endogenous GSH promotes denitrosylation, evident from increased non-protein sulfhydryl (NPSH) concentrations, predominantly composed of GSH, facilitating faster SNO reduction. The inverse correlation observed between NPSH and residual SNO levels post-chase underscores GSH’s role in maintaining nitrosothiol stability [36].

Depletion of intracellular GSH impedes protein denitrosylation, resulting in stable or elevated SNO levels. Reports suggest GSH depletion may promote *S*-nitrosylation, as GSH accepts NO from SNOs to generate GSNO, subsequently metabolized by GSH-dependent formaldehyde dehydrogenase and regulated by GSNO reductase (GSNOR) [37]. This enzyme maintains cellular GSNO and SNO equilibrium, crucial for redox homeostasis and mitigating nitrosative stress. Genetic knockout of GSNOR in mice increases cellular SNO levels, contributing to multi-organ dysfunction and heightened disease susceptibility, underscoring its role in SNO metabolism [34,37].

GSH-mediated denitrosylation mechanisms involve transnitrosylation with SNOs or direct reaction with SNOs to form *S*-glutathionylated proteins (PSSG). Despite GSH’s direct denitrosylation capabilities, evidence suggests indirect pathways, notably through GSNOR-mediated GSNO reduction. GSNOR, primarily active in the liver and cytoplasm, catalyzes GSNO reduction using NADH, generating glutathione disulfide (GSSG), which is recycled to GSH by glutathione reductase (GR) [38]. This cycle ensures effective SNO signaling modulation and protein protection from inappropriate *S*-nitrosylation events. In summary, we can say that GSH and GSNOR collaborate synergistically in regulating protein *S*-denitrosylation, critical for maintaining cellular redox balance and mitigating nitrosative stress in MS and other diseases.

Furthermore, in bacterial and mammalian cells, various transnitrosylases, such as cytochrome c, cytoglobin, caspase 3, thioredoxin, and hemoglobin’s, have been identified and functionally characterized [39,40]. However, within plant systems, GSNO is recognized as a primary transnitrosylase, playing a role in regulating the overall *S*-nitrosothiol (SNO) content [31,41]. Research indicates that the process of transnitrosylation, involving the transfer of a protein (-SNO) to another protein with an unoccupied thiol, contributes to the regulation of NO-mediated signaling pathways (Figure 3) [41]. For example, the targeted removal of nitric oxide modifications from *S*-nitrosylated proteins by Trx-h3 and Trx-h5 was discovered to govern plant immune responses in Arabidopsis [42]. In a separate investigation, it was noted that auxins in Arabidopsis roots influence the processes of *S*-nitrosylation and denitrosylation. Overall, NO-mediated *S*-nitrosylation serves as a critical signaling mechanism. Nonetheless, regulators involved in analogous processes such as transnitrosylation and denitrosylation can also adjust protein functionality in reaction to environmental stress in plants.

Previous studies have also shown that protein *S*-nitrosylation enhances the activity of ascorbate peroxidase [43]. However, it negatively regulates the activity of peroxiredoxin II E and NADPH oxidase (RBOH), thus modulating ROS signaling and oxidative stress tolerance under stressful conditions. When plants face unfavorable environmental conditions, stress-induced phytohormones have been shown to enhance their tolerance to such stresses, including sodic alkaline stress [44]. For instance, the *S*-nitrosylation of transport inhibitor response 1 (TIR1) in auxin signaling and Arabidopsis histidine phosphotransfer protein 1 (AHP1) in cytokinin signaling [45]. All these studies highlight the significance of *S*-nitrosylation in regulating various physiological processes in plants because *S*-nitrosylation is one of the most important post-translational modification mechanisms. A total of 63 and 52 *S*-nitrosylated proteins were identified in cell suspension cultures and leaves of Arabidopsis by applying exogenous NO donors. In an independent study, 926 endogenously *S*-nitrosylated proteins were identified from Arabidopsis by a site-specific nitrosoproteomics approach, which is the largest dataset of *S*-nitrosylated proteins among all organisms to date [46]. Scientists utilized RNA interference (RNAi) to generate several lines of GSNOR knockdown tomato plants, which exhibited excessive accumulation of endogenous NO and displayed phenotypes sensitive to sodic alkaline stress. Through this approach, they identified GSNOR-mediated *S*-nitrosylation, providing a more comprehensive map of the *S*-nitrosoproteome in tomato plants and offering vital insights into the molecular basis of sodic alkaline stress sensing mediated by GSNOR. Subsequently, they conducted Gene Ontology (GO) analysis to elucidate the regulatory mechanisms of *S*-nitrosylated proteins in specific biological processes in wild-type and GSNOR knockdown plants under both control and sodic alkaline stress conditions [47].

In total, 754 *S*-nitrosylated proteins were categorized according to biological processes, encompassing a broad spectrum of metabolic processes (31.96%), cellular processes (27.72%), single-organism processes (20.69%), response to stimuli (6.23%), cellular component organization or biogenesis (4.51%), biological regulation (3.98%), localization (3.18%), and six other processes (1.72%). For cellular components, 515 S-nitrosylated proteins were classified into categories such as cell (39.22%), organelle (29.13%), macromolecular complex (15.34%), membrane (11.07%), extracellular region (2.14%), and four other processes (3.11%). Additionally, 426 *S*-nitrosylated proteins were categorized by molecular function, including catalytic activity (41.77%), binding (40.78%), structural molecule activity (8.66%), transporter activity (3.46%), antioxidant activity (1.95%), molecular function regulation (1.41%), electron carrier activity (1.17%), and four other functions (1.41%) [48].

Moreover, 47 *S*-nitrosylated proteins with 53 *S*-nitrosylated sites have been identified in the response to stimulus. Among these proteins, only one protein, calcium sensing receptor (CAS; K4BKU7), involving stress response signaling has been identified, which implies that the crosstalk between RNS and Ca ^2+^ signaling in plants stress tolerance may depend on these post-translational modifications. As a common toxic metabolite, ROS bust can be detected nearly under all stress conditions. Moreover, nine ROS-scavenging enzymes, including two superoxide dismutases (SOD; Q9SBJ4 and Q7XAV2), two catalases (CAT; P30264 and Q9XHH3); two ascorbate peroxidases (APX; Q3I5C4 and Q09Y77), one dehydroascorbate reductase (DHAR; Q4VDN8), and two thioredoxins (TRX; K4BVS6 and K4DCR6) were identified as *S*-nitrosylated proteins, supporting the notion of active interactions between RNS- and ROS- mediated signaling pathways in stress tolerance. In addition, three key enzymes of NO metabolism, GSNOR (D2Y3F4), nitrite reductase (NIR; K4B378) and bifunctional nitrilase/nitrile hydratase (NIT; K4DA30), have also been identified in this *S*-nitrosoproteome, which have been consistent with the results of biotin-switch assay in previous study [49] suggesting the involvement of feedback regulation in NO-mediated *S*-nitrosylation. Collectively, these data suggest a mechanism for NO signal transduction in which GSNOR nitrosation and inhibition transiently permit GSNO accumulation. Additionally, GO subcellular location analysis revealed that the *S*-nitrosylated proteins were enriched in peroxisomes for 1.20% [50] Moreover, advancements in methodologies have progressed, and the catalog of plant peroxisomal proteins susceptible to undergoing specific post-translational modifications (PTMs) derived from nitric oxide (NO) has expanded. The characteristic peroxisomal proteins identified as targets of NO in higher plants are outlined in Table 1. Within the scope of this article, attention is primarily directed towards several key antioxidant enzymes located within peroxisomes, notably CAT, MDAR, and CuZnSOD. In the following paragraphs, we will delve deeper into this process.

### 4.1. Catalase

Catalase (CAT; EC 1.11.1.6) is predominantly a tetrameric protein containing iron. Its primary role is to catalyze the dismutation of H_2_O_2_ into H_2_O and molecular O_2_. Catalase occupies a central position in the antioxidative metabolic framework of both prokaryotic and eukaryotic cells [58,59,60]. Due to its kinetic properties, catalase efficiently controls H_2_O_2_ levels within specific ranges. As a result, it is acknowledged as a key antioxidant enzyme, crucially involved in diverse developmental stages and responses to stress conditions. [59]. In living organisms, the elimination of H_2_O_2_ relies on several enzymatic systems. In animal cells, in addition to CAT, both selenium-dependent and selenium-independent glutathione peroxidases (GPX; EC 1.11.1.9) also play roles in reducing this ROS to levels essential for physiological functions [60]. The necessity to maintain the function of selenium-dependent glutathione peroxidases (Se-GPXs) highlights the dietary dependence on selenium. Furthermore, specific peroxiredoxins (Prxs), identified as small proteins with peroxidase activity, also contribute to managing elevated levels of intracellular H_2_O_2_ [61]. The catalase system, recognized as one of the earliest and first identified antioxidant enzymes, was initially documented by O. Loew in 1900. Through subsequent research, it has been established that catalase not only functions as a powerful antioxidant but also serves as a crucial element of cellular metabolism in all aerobic organisms. For example, studies have shown that human catalase displays 245 single-nucleotide polymorphisms, influencing a range of physiological and pathological conditions [62]. 

Besides genetic factors, catalase activity can be influenced by variables such as age, physical activity, seasonal changes, and specific chemicals. Catalase is also implicated in various physiological conditions including hypertension, diabetes mellitus, insulin resistance, dyslipidemia, asthma, bone metabolism, and vitiligo. However, in specific physiological contexts, catalase plays a central role in processes where H_2_O_2_ and other ROS act as signaling molecules, requiring precise regulation for appropriate cellular responses [63].

Recently, there has been growing attention towards other PTMs induced by RNS, primarily due to their intricate and complex mechanisms. RNS originate from NO, a radical gas with significant signaling importance. Among the three PTMs induced by RNS—S-nitrosation, tyrosine nitration, and metal nitrosylation-the former two have received considerable focus. Both terms “S-nitrosylation” and “S-nitrosation” have been used to describe this PTM [64].

### 4.2. S-Nitrosation of Catalase

Concerning S-nitrosation, unbound NO can directly bind to the thiol group of cysteine, leading to the production of a group of compounds known as *S*-nitrosothiols. Among these, one of the most significant *S*-nitrosothiols is generated through the interaction of NO with reduced glutathione, resulting in the formation of GSNO. GSNO possesses the capability to release NO, thus serving as a potential donor for further *S*-nitrosation reactions. This characteristic confers upon GSNO a role as a reservoir for NO, facilitating its involvement in *S*-nitrosation processes [65]. In a research endeavor conducted on leaf specimens from the plant model Arabidopsis thaliana, the identification of cysteine residues exhibiting heightened susceptibility to *S*-nitrosation was achieved. This investigation utilized GSNO as a NO donor for such assessments, supplemented by subsequent computational analysis. Notably, prior findings revealed that GSNO exerted an inhibitory effect on the catalase activity of A. thaliana leaves, resulting in a reduction of up to 25–30% in an extent contingent upon concentration. To further understand how NO modulates catalase activity, a computational investigation was conducted. This involved docking GSNO to the quaternary structure of Arabidopsis catalase isozymes CAT-1, CAT-2, and CAT-3, as estimated with a special technique.

Despite the high similarity, the alignment of their primary structures revealed that the positions of the various Cys residues are not entirely conserved [66]. Subsequent in silico investigations involved computational modeling of the quaternary structure of Arabidopsis catalases. Blind docking of GSNO with the models of the three catalases resulted in various poses, which were ranked based on the full fitness parameter [67]. When considering the distance between the sulfur (S) atoms of GSNO and cysteine (Cys) as a discriminant criterion, the number of solutions was reduced to two poses with varying estimated affinity for each isoenzyme. Notably, Cys420 emerged as a common putative *S*-nitrosation target across all catalases.

The site proximal to Cys420 was denoted as S1, while the others were labeled as S2. Based on the computed ΔG values, the dissociation constant (Kd) of the GSNO-catalase interaction was estimated. The relative affinity values suggest that GSNO affinity is approximately one-fold higher for site 1 of CAT-2 and slightly elevated for site 2 in CAT-1. Remarkably, CAT-1 has been recognized as a significant contributor to H_2_O_2_ elimination under various environmental stresses, and the role of glutathione status in conveying signals originating from intracellular H_2_O_2_ has been postulated [68].

### 4.3. Tyrosine Nitration and Metabolism of RO

Stress conditions that raise NO and ABA levels are usually accompanied by ROS such as superoxide anion, hydroxyl radicals, and hydrogen peroxide, which together are involved in multiple signaling pathways [69]. The levels of ROS are controlled by the activity of antioxidant enzymes, mainly superoxide dismutase and catalase that metabolize superoxide and hydrogen peroxide, respectively. These enzymes are regulated by NO-related PTMs [70]. Nitration and *S*-nitrosylation of catalases have been reported. Regarding CAT, the enzymes of pepper fruits were inhibited by Tyr nitration during ripening though specific target residues have not yet been identified. Hydrogen peroxide levels are also regulated through the so-called ascorbate glutathione antioxidant cycle involving the activity of APX, MDAR, and DHAR, and glutathione reductases [71]. As for SODs and CATs, also enzymes of the ascorbate glutathione cycle have been identified as nitration targets. The Tyr nitration sites for pea APX and MDAR have been identified [72]. DHAR and GR have also been identified as Tyr nitration targets in citrus and sunflower respectively, but nitration sites remain to be determined. It is worth mentioning that nitration of GR did not affect their activity, thus representing one of the few cases where nitration did not lead to enzyme inactivation. The rest of antioxidant enzymes are inhibited by Tyr nitration thus representing a very relevant node of regulation of ROS metabolism by NO (Figure 4).

### 4.4. Monodehydroascorbate Reductase

This enzyme is a component of the ascorbate-glutathione cycle, which serves the additional role of regulating the cellular levels of H_2_O_2_ [73]. The ASC-GSH cycle is distributed across various subcellular compartments, including peroxisomes. Furthermore, extensive proteomic investigations have identified GR and MDAR as crucial components of the ascorbate–glutathione cycle that undergo nitration and *S*-nitrosylation [72,73,74]. These studies aim to elucidate the molecular mechanisms and physiological relevance of NO-derived PTMs in regulating the activities of MDAR and GR, essential enzymes in maintaining cellular redox homeostasis.

GR is pivotal in the antioxidative defense system, facilitating the conversion of oxidized glutathione (GSSG) to GSH using NADPH as a cofactor. This process is crucial for maintaining a high GSH/GSSG ratio, given the prominent role of GSH as the most abundant soluble antioxidant in plants [75,76]. Under experimental conditions, the activities of GR isoforms were unaffected by NO-PTMs induced by ONOO^–^ and GSNO, suggesting a potential mechanism for preserving GSH regeneration and sustaining antioxidant capacity against nitro-oxidative stress. Notably, the resistance of pea GR to nitration by peroxynitrite is a rare observation in higher plants, where nitration typically results in loss of protein function [77].

With pea GR showing resilience to NO-PTMs, focus shifted to MDAR, an enzyme critical for regenerating reduced ascorbate within the ascorbate glutathione cycle. MDAR in pea plants is encoded by a single gene, with its protein localized across various subcellular compartments. MDAR plays a significant role in responding to environmental stresses characterized by nitro-oxidative conditions. While MDAR activity in pea increases under high light intensity and cadmium stress, it decreases during natural leaf senescence and under the herbicide 2,4-D. Similar stress-responsive patterns are observed in other plants such as tomato, rice, and Arabidopsis, where MDAR activity fluctuates under diverse environmental stresses [78].

Despite MDAR being identified as a target for nitration and *S*-nitrosylation in proteomic studies, the specific consequences of these PTMs on MDAR function remained unclear until recently. It has been shown that both nitration and *S*-nitrosylation lead to a loss of MDAR function. While three tyrosine residues have been implicated in nitration, and two cysteine residues are present in pea MDAR, their exact roles in functional inhibition are complex to ascertain due to their positions not directly affecting functionality. However, the evolutionary conservation of these residues suggests their critical reactivity in response to NO-PTMs [79].

The radical mechanism of tyrosine nitration involves indirect reactions with peroxynitrite, facilitated by carbon dioxide or metal centers, resulting in the formation of a tyrosyl radical. Based on interaction network analyses with cofactors, Tyr345 emerges as a significant site for nitration in MDAR, influencing its functionality through interactions with FAD atoms N3 and N10. Further computational analyses using PropKa 3.1 highlighted extreme pKa values for these atoms, supporting the hypothesis that nitration at Tyr345 disrupts MDAR functionality. This hypothesis was validated through site-directed mutagenesis, where the Tyr345Phe mutant showed resilience to ONOO^–^, confirming the role of Tyr345 in regulating MDAR activity [72,73,74,75,76,77,78,79,80].

Furthermore, MDAR activity is regulated by GSNO, suggesting a comprehensive regulatory network integrating MDAR with NO metabolism and iron metabolism via plant oxyhemoglobin (Hb). The presence of NO-related species including ONOO– and GSNO in peroxisomes, alongside _L_-arginine-dependent nitric oxide synthase activity, further underscores the potential modulation of peroxisomal MDAR under oxidative stress conditions [81]. Notably, under salt stress-induced nitro-oxidative conditions, MDAR expression, mRNA levels, protein abundance, and activity were found to increase, indicating a compensatory mechanism to counteract inhibitory NO-PTMs in pea leaves.

Overall, the interplay between nitration and *S*-nitrosylation, mediated by NO-derived molecules, plays a pivotal role in regulating the activities of key antioxidative enzymes such as GR and MDAR in plants. These PTMs not only influence enzymatic function but also integrate MDAR into broader physiological responses to environmental stresses. Future research should explore the intricate molecular mechanisms underlying NO-PTMs, particularly their impact on protein structure and function, to enhance our understanding of plant adaptation and stress tolerance mechanisms mediated by NO signaling pathways.

However, limited data exist regarding how NO can modulate the specific isozymes of this cycle found within peroxisomes.

### 4.5. SOD Emerges as a Significant Protein Warranting Further Exploration as a Target of PTMs Mediated by NO

SODs are a group of metalloenzymes that facilitate the conversion of superoxide radicals O_2_^•−^ into H_2_O_2_ and oxygen (O_2_) via disproportionation. In higher plants, three predominant types of SODs are identified, characterized by their prosthetic metal ions: manganese (Mn-SODs), Fe-SODs, or a combination of CuZn-SODs. The presence of various forms of SODs within plant peroxisomes has been documented in a minimum of ten diverse plant species [68,69,70,71,72,73,74,75,76,77,78,79,80,81,82]. Currently, SOD is recognized as a consistent enzyme present in all categories of peroxisomes, although the specific array of isozymes varies depending on the organ and species of the plant. Regarding the susceptibility of SOD to various modifications induced by RNS, earlier studies suggested that recombinant human Mn-SOD and CuZn-SOD were susceptible to inactivation by ONOO^−^ [83]. Recently, recombinant peroxisomal CuZn-SOD was successfully produced in Arabidopsis. In vitro experiments conducted in the presence of nitrating or *S*-nitrosylating agents revealed that ONOO^−^ led to a reduction in Cu,Zn-SOD activity, while GSNO did not induce any observable effect. Mass spectrometric analyses identified Tyr115 as the likely target of nitration [84]. Therefore, SOD emerges as a significant protein warranting further exploration as a target of PTMs mediated by NO, given its sensitivity that suggests a potential ability to differentiate between nitration and nitrosation processes.

## 5. Conclusions and Future Perspectives

Our current grasp of how peroxisomes contribute to redox biochemistry and regulate the transcriptome in response to environmental changes remains incomplete, highlighting the need for several key areas of future exploration. To advance this understanding, comprehensive characterization of the peroxisomal reactive species interactome is essential. This includes identifying the production of hydrogen sulfide (H_2_S) within peroxisomes, mapping the sulfenylome and persulfidome of peroxisomal proteins, and elucidating the interactions among all reactive species and their targets. It is also crucial to examine the role of catalase (CAT) in GSNOR-mediated transnitrosylation, which raises the possibility that other proteins, both within and outside of peroxisomes, could undergo similar modifications, as well as to explore whether other transnitrosylases exist within peroxisomes. Understanding the sources of nitric oxide (NO) within peroxisomes and identifying NO-dependent transcriptomic changes are among the most challenging aspects of peroxisome and redox biology in plants. Furthermore, discovering peroxisomal redox relays involved in retrograde signaling and studying carbonylated and nitrated peroxisomal proteins as signaling messengers are vital areas for deeper investigation to grasp how peroxisomes detect, translate, and regulate cellular responses to environmental shifts.

Looking ahead, research should focus on the diverse roles of NO in seed germination and abscisic acid (ABA) signaling to uncover the precise molecular mechanisms through which NO influences these critical processes. Understanding how NO interacts with other key signaling molecules, such as gibberellins and calcium ions, is crucial for grasping how NO regulates seed dormancy and germination across varying environmental conditions. Further studies into NO’s effects on ABA catabolic gene expression and its involvement in post-translational modifications may identify new targets for enhancing crop resilience to abiotic stresses. Additionally, examining the dual and context-dependent effects of NO on ABA signaling pathways, especially concerning stomatal movement and stress responses, could reveal innovative approaches to improve plant stress tolerance and productivity. Progress in high-resolution proteomics, real-time imaging, and advanced genetic tools will be vital for elucidating NO’s complex role and optimizing its use in agricultural practices.

## Figures and Tables

**Figure 1 ijms-25-08873-f001:**
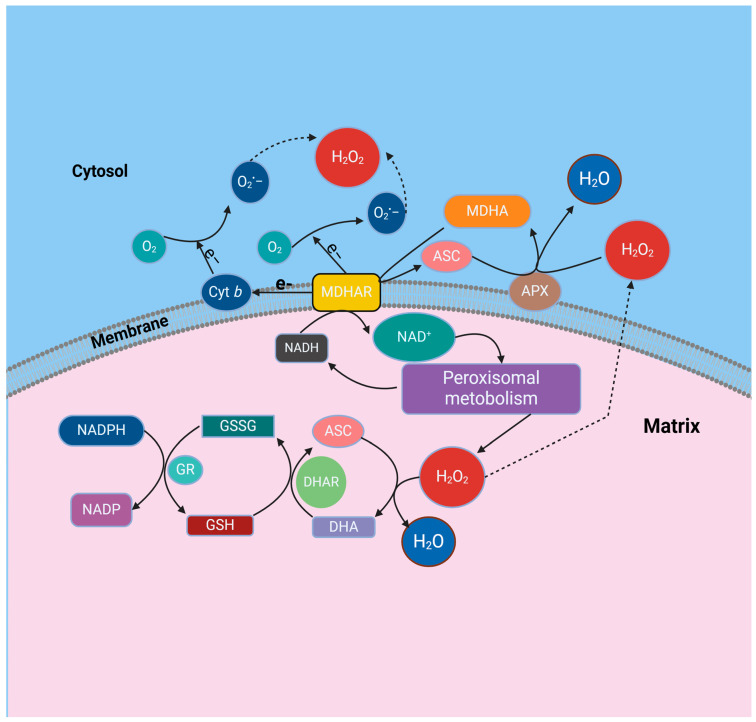
A proposed model for the function of the ascorbate glutathione cycle in leaf peroxisomes is derived from various experimental observations. These include studies on enzyme activity latency within intact organelles, solubilization assays using 0.2 M KCl, characterization of peroxisomal membrane polypeptides (PMPs) from pea leaves, and investigations into the NADH-dependent monodehydroascorbate reductase (MDHAR) activity in peroxisomal membranes from castor bean endosperm. The model integrates components such as ascorbate (ASC) in its reduced (ASC) and oxidized (dehydroascorbate, DHA) forms, monodehydroascorbate reductase (MDHAR), glutathione reductase (GR), reduced glutathione (GSH), oxidized glutathione (GSSG), ascorbate peroxidase (APX), and xanthine oxidase (XOD).

**Figure 2 ijms-25-08873-f002:**
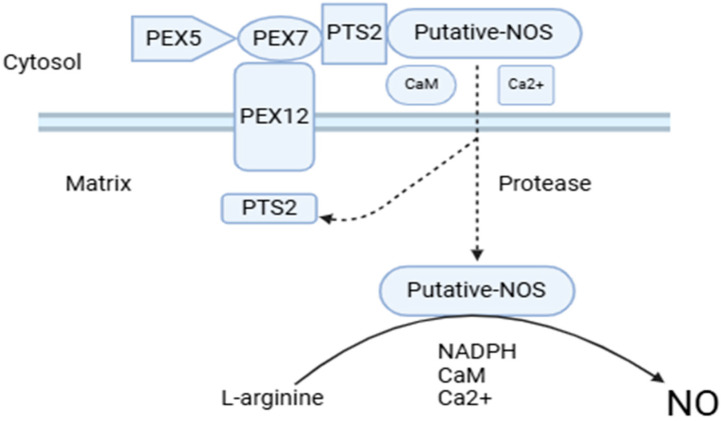
The key components involved in the process of importing the peroxisomal protein that is responsible for generating nitric oxide (NO) into the peroxisome.

**Figure 3 ijms-25-08873-f003:**
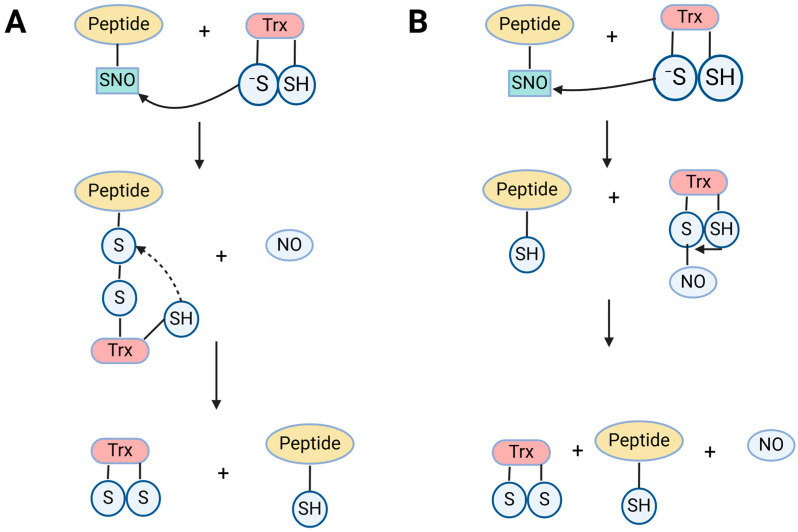
Outline of the nitric oxide (NO)-mediated post-translational modifications alternative reactions by Trx. Proteins are represented with letter “P.” (**A**) The initial stage involves a nucleophilic attack on the SNO by the catalytic cysteine, which releases nitric oxide (NO) and creates a mixed disulfide intermediate. This intermediate is then reduced by the resolving thiol of Trx. (**B**) In the alternative reaction, the process begins with the same initial step, where NO is transferred to the catalytic thiol, resulting in the release of the reduced peptide. Following this, the bound SNO is reduced by the resolving cysteine, which subsequently releases NO.

**Figure 4 ijms-25-08873-f004:**
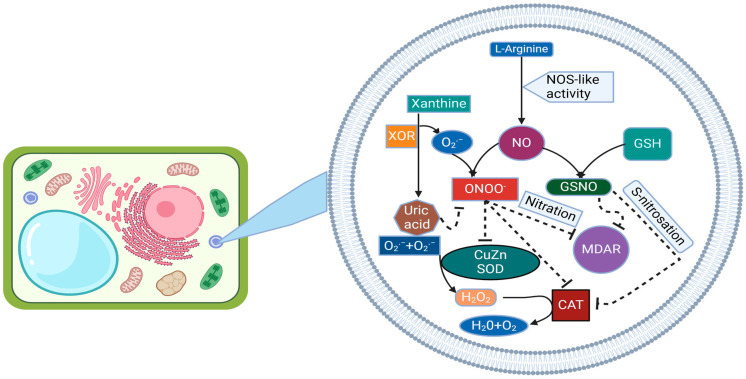
The intricate interplay between nitric oxide (NO) metabolism and antioxidant enzymes within plant peroxisomes involves several key components. Peroxisomal xanthine oxidoreductase (XOR) activity generates uric acid and superoxide radicals (O_2_^•−^), while an _L_-arginine-dependent nitric oxide synthase (NOS)-like activity produces NO. NO can react with O_2_^•−^ to form ONOO^−^, a potent oxidant capable of inducing PTMs like tyrosine nitration. Additionally, NO can combine with GSH to yield GSNO, serving as a NO donor for *S*-nitrosation reactions. Uric acid, known for its ability to scavenge ONOO^−^, may contribute to a regulatory mechanism within peroxisomes. These interactions influence the activity of key peroxisomal enzymes, including CAT, CuZnSOD, and MDAR, potentially leading to their inhibition.

**Table 1 ijms-25-08873-t001:** Certain proteins originating from peroxisomes in higher plants experience post-translational modifications (PTMs) derived from nitric oxide (NO), including *S*-nitrosation or tyrosine nitration and affect their activities.

Peroxisomal Enzyme	Function	Effect on Activity	Effect on Activity	References
Hydroxypyruvate reductase (HPR)	Photorespiration	*S*-nitrosation/Tyr nitration	Inhibition/Inhibition	[51]
Glycolate oxidase (GOX)	Photorespiration	*S*-nitrosation/Tyr nitration	Inhibition/Inhibition	[52]
Malate dehydrogenase (MDH)	Fatty acid β-oxidation	*S*-nitrosation/Tyr nitration	Inhibition/Inhibition	[53]
Catalase (CAT)	H_2_O_2_ decomposition	*S*-nitrosation/Tyr nitration	Inhibition/Inhibition	[54]
CuZn superoxide dismutase (CSD3)	O2^•−^ dismutation	Tyr nitration	Inhibition/Inhibition	[55]
Monodehydroascorbate reductase (MDAR)	Ascorbate glutathione cycle	*S*-nitrosation/Tyr nitration	Inhibition/Inhibition	[56,57]

## Data Availability

Not applicable.

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
