# Peer review of "The Key Targets of NO-Mediated Post-Translation Modification (PTM) Highlighting the Dynamic Metabolism of ROS and RNS in Peroxisomes"

_ijms, 2024, doi:10.3390/ijms25168873_

Round 1

Reviewer 1 Report

Comments and Suggestions for Authors

The manuscript by Ergashev et al. try to focus on specific targets of NO-PTMs in peroxisomes in plants, especially in those aspects that interconnects ROS and RNS metabolism. Unfortunately, I did not come away from reading it with a clear idea of what is known and not known about the topic.

In any case, I believe this revision could be of interest and can be considerably improved.

My comments to improve the manuscript:

1) The title of the manuscript suggests that the authors will focus on the regulation of specific peroxisome targets by NO-PTMs; however, much of the text focus on general descriptions about how NO-PTMs are produced and other general information. For instance, there are one page focused on the de-nitrosylation process, but not related to peroxisome activity. Late in the manuscript (page 9) the authors start to mention the NO-PTMs in peroxisomes that unfortunately are not in-depth covered. In this regard, authors should extend the information of the specific targets of NO-PTMs in this organelle and its potential consequences under physiological and/or stress conditions. Maybe, there is no so much information on the antioxidant enzymes modulated by NO in peroxisomes; in this case authors should reconsider the objective of the review and maybe extend the discussion to all NO-targets in peroxisomes, even including animal models, when possible.

Please, note that this single comment involves almost a rewriting of the manuscript.

2)  The authors should review the bibliography in depth as there are some errors throughout the manuscript that have made reading difficult. I highlight some of them (although not all):

a. L204: ‘….S-nitrosylation proteins by Trx-h3 and Trx-h5 was discovered to govern plant   immune responses in Arabidopsis (42)’. This ‘42’ reference investigates on Trxh-2 but does not talk about Thrx-h3 and 5 and its role in de-nitrosylation and plant immunity. I guess authors could be discussing the paper published by Kneeshaw et al. in Molecular Cell in 2014 or another one.

b. L227: ‘….nitrosoproteomics approach, which is the largest dataset of S-nitrosylated proteins among all organisms to date (28)’. This reference is a review that does not identify S-nitrosylated proteins in Arabidopsis. Maybe authors wanted to reference here the number 63 of their reference list that shows a list of S-nitrosylated proteins in Arabidopsis.

c. L263. the reference 60 is about tyrosine nitration, not S-nitrosylation.

d.    L334 and L338, references 50 and 51. These references do not show S-nitrosylation and structural characterization of the catalase isoforms. Again, I guess authors maybe are discussing the paper of Palma et al. published in Redox Biology in 2020 that has similar results or maybe another one.

e.   Please, revise all the references in the text and the reference list.

3) L105-L110. MDAR is an antioxidant enzyme involved in the Asc-GSH cycle thanks to its capacity to regenerate reduced ascorbate. However, in these lines is proposed that MDAR can mediate the production of superoxide anion that is just the opposite to be antioxidant. Please, describe this in more detail and provide references about this MDAR function.

4) L126 and L131. Authors mention a NOS enzyme, but this enzyme has not been identified in plants. For this reason, it should appear NOS-like activity if they are referring to plants. In the case they are talking about animal models, they should clearly specify it.

5) L144-L146. This part should be rephrased because it sounds like there is no tyrosine nitration under physiological conditions. This modification is a nitrosative stress marker but it also occurs under non-stress conditions as different papers have shown the physiological nitration of different target proteins.

6)  L171 Please replace nitrosothiol by nitric oxide since S-nitrosylation consist of the addition of NO to an SH forming a nitrosothiol.

7) L197. Add a reference on transnitrosylates in mammals.

8) Table 1. I believe the manuscript would benefit from a more detailed table. Authors could provide, when known, the effect of NO-PTMs on the protein function, the residue modified by these modifications, the plant specie in which the modification was identified and finally the reference where they obtained the information.

9) Line 39: The abbreviation ROS and RNS are already introduced. This is repeatedly over the text with all abbreviations. Please revise the manuscript and introduce the abbreviation and full name just the first time in the text and then use only the abbreviation.

 10) .   L114 and L201: Please remove Look at when citing a figure in the text.

11) Please revise the abbreviation of all reactive species and correct the superscripts and/or subscripts.

12)  FIGURES

a.  Figure 1. In the matrix, what enzyme is converting H2O2 into H2O? is it APX? The oxidation of Asc could led to MDA formation and then ascorbate is regenerated by MDAR function. Is it not possible in the matrix? Authors only show the DHAR activity.

b.    In my opinion figure 3 could be removed since it does not offer relevant information on NO-PTMs in peroxisome.

13) It would be good if the authors can provide a clear message of what should be investigated in this area to provide more useful information about NO-PTMs in peroxisomes. They comment that it is interesting to investigate if GSNOR is able to regulate total cellular S-nitrosylation levels under abiotic stress. However, it is well known that this already happens under different stress conditions: wounding, high temperature, salt, among others. Consequently, the authors should clarify what aspects should be investigated in this topic. In the same line, they state in L436 that ‘investigating the role of S-nitrosylation in ABA function could be a valuable and significant area of study’. In my opinion, authors should clarify what is interesting to investigate in this ABA-SNOs relation since there are works in the literature on how S-nitrosylation of proteins can modify ABA signalling, specially related to seed dormancy and germination.

Author Response

We thank both reviewers very much for the constructive comments and suggestions on improving our manuscript. We have carefully considered the comments and suggestions and revised the manuscript to address them. Please find our point-by-point responses below. The paragraphs in blue were our responses. Revised contents in the manuscript were marked in yellow.

Reviewer #1 (Remarks to the Author): The manuscript by Ergashev et al. try to focus on specific targets of NO-PTMs in peroxisomes in plants, especially in those aspects that interconnects ROS and RNS metabolism. Unfortunately, I did not come away from reading it with a clear idea of what is known and not known about the topic. In any case, I believe this revision could be of interest and can be considerably improved

My comments to improve the manuscript:

1) The title of the manuscript suggests that the authors will focus on the regulation of specific peroxisome targets by NO-PTMs; however, much of the text focus on general descriptions about how NO-PTMs are produced and other general information. For instance, there are one page focused on the de-nitrosylation process, but not related to peroxisome activity. Late in the manuscript (page 9) the authors start to mention the NO-PTMs in peroxisomes that unfortunately are not in-depth covered. In this regard, authors should extend the information of the specific targets of NO-PTMs in this organelle and its potential consequences under physiological and/or stress conditions Maybe, there is no so much information on the antioxidant enzymes modulated by NO in peroxisomes; in this case authors should reconsider the objective of the review and maybe extend the discussion to all NO-targets in peroxisomes, even including animal models, when possible.

Please, note that this single comment involves almost a rewriting of the manuscript.

Answer: We agree with the reviewer’s comment. We have rewritten more deeply and understandable some page contents in order to improve the clarity and coherence (See the lines 146-154, 177-181 and 195-225 in the revise manuscript text).

The passage Lines 138-146 underscores the close relationship between ROS and NO-related modifications, the reversible nature of S-nitrosation compared to irreversible nitration, and the specific targeting of peroxisomal proteins, including crucial ones like LON2 protease, by NO-mediated PTMs.

Furthermore, we rewrite content about de-nitrosylation with deeper understanding about the importance of GSH and GSNOR in regulating protein S-nitrosylation and denitrosylation. GSH’s role in maintaining redox balance and mitigating nitrosative stress is crucial, especially in the context of diseases like multiple sclerosis, where these mechanisms are disrupted. (See the lines 206-236 in the revise manuscript text)

At the beginning of the paragraph “Nitric Oxide - Induced S-Nitrosylation, Tyrosine nitration, Transnitrosylation, and Denitrosylation Processes” we highlighted about the critical role of NO-mediated S-nitrosylation, its regulation by GSH and GSNOR, and it impact on redox balance, signaling pathways, and stress tolerance across different organisms, in order later to explain in detail about the text emphasizes the complexity of NO-mediated post-translational modifications and their critical roles in maintaining cellular function and adapting to stress. It highlights the intricate balance between S-nitrosylation and its regulatory mechanisms, including the roles of GSH and GSNOR, in various biological contexts and diseases both plant and animal organisms (See the lines 177-236 in the revise manuscript text).

2)  The authors should review the bibliography in depth as there are some errors throughout the manuscript that have made reading difficult. I highlight some of them (although not all):

  1. L204: ‘….S-nitrosylation proteins by Trx-h3 and Trx-h5 was discovered to govern plant   immune responses in Arabidopsis (42)’. This ‘42’ reference investigates on Trxh-2 but does not talk about Thrx-h3 and 5 and its role in de-nitrosylation and plant immunity. I guess authors could be discussing the paper published by Kneeshaw et al. in Molecular Cell in 2014 or another one.

Answer: Thank you for your special attention to such details, errors in the references have been corrected.

 In the line 245 in the revise manuscript text we described like example the enzymes Trx-h3 and Trx-h5 are involved in removing these NO modifications from proteins (a process called denitrosylation) and this removal process is crucial for regulating plant immune responses in Arabidopsis. Essentially, Trx-h3 and Trx-h5 help control how proteins involved in immune responses are modified and thus affect how the plant responds to threats.

  1. L227: ‘….nitrosoproteomics approach, which is the largest dataset of S-nitrosylated proteins among all organisms to date (28)’. This reference is a review that does not identify S-nitrosylated proteins in Arabidopsis. Maybe authors wanted to reference here the number 63 of their reference list that shows a list of S-nitrosylated proteins in Arabidopsis.

Answer: Yes, it has been removed and revised to Ferrer-Sueta, G.; Campolo, N.; Trujillo, M.; Bartesaghi, S.; Carballal, S.; Romero, N.; Alvarez, B.; Radi, R. Biochemistry of peroxynitrite and protein tyrosine nitration. Chem. Rev. 2018, 118, 1338–1408 doi: 10.1021/acs.chemrev.7b00568

  1. L263. the reference 60 is about tyrosine nitration, not S-nitrosylation.

Answer: Revised (Line 781-783 in the revise manuscript text)

  1. L334 and L338, references 50 and 51. These references do not show S-nitrosylation and structural characterization of the catalase isoforms. Again, I guess authors maybe are discussing the paper of Palma et al. published in Redox Biology in 2020 that has similar results or maybe another one.

Answer: Yes, references have corrected

  1. Please, revise all the references in the text and the reference list.

Answer: Thanks for the suggestion of the reviewer, we check and corrected the sequence of numbers through all reference list.

13) It would be good if the authors can provide a clear message of what should be investigated in this area to provide more useful information about NO-PTMs in peroxisomes. They comment that it is interesting to investigate if GSNOR is able to regulate total cellular S-nitrosylation levels under abiotic stress. However, it is well known that this already happens under different stress conditions: wounding, high temperature, salt, among others. Consequently, the authors should clarify what aspects should be investigated in this topic. In the same line, they state in L436 that ‘investigating the role of S-nitrosylation in ABA function could be a valuable and significant area of study’. In my opinion, authors should clarify what is interesting to investigate in this ABA-SNOs relation since there are works in the literature on how S-nitrosylation of proteins can modify ABA signalling, specially related to seed dormancy and germination.

Answer: Thanks a lot, we agree with your suggestion and we have added the paragraph about “Role of Nitric Oxide (NO) in Seed Germination and Abscisic Acid (ABA) Signaling” which overall, the paragraph highlights NO’s dual and complex role in regulating seed germination and various physiological processes by interacting with ABA and other signaling molecule (See the lines 425-468 in the revise manuscript text).

Furthermore, we rewrite the Conclusion and future perspectives with outlines key research areas needed to deepen our understanding of peroxisomes and nitric oxide (NO) in plants. It calls for a detailed analysis of peroxisomal reactive species, including hydrogen sulfide (H2S), and their interactions with peroxisomal proteins, as well as the exploration of catalase (CAT) in transnitrosylation processes. The study should also focus on identifying NO sources within peroxisomes, understanding NO's impact on gene expression, and investigating peroxisomal redox relays and signaling mechanisms. Additionally, it stresses the importance of exploring NO's roles in seed germination and abscisic acid (ABA) signaling, including its interactions with other signaling molecules and its effects on stress responses. Advancements in proteomics, imaging, and genetic tools are deemed crucial for unraveling these complex processes and enhancing plant resilience and productivity. (Seethe lines 553-582 in the revise manuscript text)

Reviewer 2 Report

Comments and Suggestions for Authors

The manuscript entitled “The key targets of NO-mediated post-translation modification (PTM): highlighting the dynamic metabolism of ROS and RNS in peroxisome” offers a comprehensive overview of the intricate relationship between reactive oxygen species (ROS) and reactive nitrogen species (RNS) metabolism within peroxisomes. The authors have effectively summarized the pivotal regulatory role of Nitric oxide (NO) in the intricate interaction between enzymes and their activities. They highlighted how NO modulates these activities through specific post-translational modifications (PTMs), with a particular focus on S-nitrosylation and tyrosine nitration, which are key mechanisms in the dynamic metabolism of reactive oxygen and nitrogen species within peroxisomes. However, the manuscript requires significant improvements before it can be considered for publication. I recommend that the authors undertake major revisions to address the following concerns:

Major comments:

1. The manuscript extensively delineates the mechanisms underpinning the action of reactive oxygen species (ROS); however, it appears that the functional importance of reactive nitrogen species (RNS) is not adequately highlighted. Given the pivotal role of RNS in diverse plant physiological processes and stress responses, it is advisable to integrate a more in-depth exploration of RNS dynamics into the manuscript.

2. To enhance the comprehensibility of the manuscript, it is advisable to incorporate diagrams that succinctly encapsulate the key information presented in sections located at lines 142-143 and 168. The current density of information in these sections runs the risk of overwhelming readers, potentially compromising the overall accessibility of the text. Visual representations, such as diagrams or schematic illustrations, would effectively distill the complexity, thereby facilitating a clearer understanding of the material.

3. Numerous sentences throughout the text are devoid of accompanying references. The inclusion of proper citations is essential for bolstering the credibility of the work and integrating it within the wider scholarly discourse. The authors should meticulously review the manuscript to pinpoint all instances where citations are necessary and verify that they are buttressed by pertinent and current references. For instance, refer to lines 142-143 and 195-196.

Maior comments:

1. Please provide elucidation regarding the discrepancies depicted between panels A and B.

2. In Line 300, the sentence should conclude with a full stop (period) to adhere to standard punctuation conventions.

3. Please provide a brief description of the function of each enzyme listed in Table 1. This will offer readers a clear understanding of the role these enzymes play in the context of the study.

Author Response

Reviewer #2 (Remarks to the Author): The manuscript entitled “The key targets of NO-mediated post-translation modification (PTM): highlighting the dynamic metabolism of ROS and RNS in peroxisome” offers a comprehensive overview of the intricate relationship between reactive oxygen species (ROS) and reactive nitrogen species (RNS) metabolism within peroxisomes. The authors have effectively summarized the pivotal regulatory role of Nitric oxide (NO) in the intricate interaction between enzymes and their activities. They highlighted how NO modulates these activities through specific post-translational modifications (PTMs), with a particular focus on S-nitrosylation and tyrosine nitration, which are key mechanisms in the dynamic metabolism of reactive oxygen and nitrogen species within peroxisomes. However, the manuscript requires significant improvements before it can be considered for publication. I recommend that the authors undertake major revisions to address the following concerns:

Major comments:

  1. The manuscript extensively delineates the mechanisms underpinning the action of reactive oxygen species (ROS); however, it appears that the functional importance of reactive nitrogen species (RNS) is not adequately highlighted. Given the pivotal role of RNS in diverse plant physiological processes and stress responses, it is advisable to integrate a more in-depth exploration of RNS dynamics into the manuscript.

Answer: Many thanks for your suggestion about to give pivotal role of RNS in manuscript, considering your suggestion worthy and useful we added some points about RNS and its diverse plant physiological processes (see the lines 47-58 in the revise manuscript text). In manuscript, we tried to open interconnection between ROS and RNS, that’s why some point seem more detailed about ROS, however whole focus continuing to play an important role RNS family, NO.

  1. To enhance the comprehensibility of the manuscript, it is advisable to incorporate diagrams that succinctly encapsulate the key information presented in sections located at lines 142-143 and 168. The current density of information in these sections runs the risk of overwhelming readers, potentially compromising the overall accessibility of the text. Visual representations, such as diagrams or schematic illustrations, would effectively distill the complexity, thereby facilitating a clearer understanding of the material.

Answer: We agree with your suggestion and we made the figure which illustrates the key components involved in the process of importing the peroxisomal protein that is responsible for generating nitric oxide (NO) into the peroxisome (See the lines 171-173 in the revise manuscript text).      

  1. Numerous sentences throughout the text are devoid of accompanying references. The inclusion of proper citations is essential for bolstering the credibility of the work and integrating it within the wider scholarly discourse. The authors should meticulously review the manuscript to pinpoint all instances where citations are necessary and verify that they are buttressed by pertinent and current references. For instance, refer to lines 142-143 and 195-196.

Answer: Good suggestion! We the big content in manuscript all main texts have cited.

Maior comments:

  1. Please provide elucidation regarding the discrepancies depicted between panels A and B.

Answer: Thank you for your suggestion we made explanation for two panels Figure 2. Outline of the nitric oxide (NO)-mediated post-translational modifications alternative reactions by Trx. Proteins are represented with letter “P.” (A) The initial stage involves a nucleophilic attack on the SNO by the catalytic cysteine, which releases nitric oxide (NO) and creates a mixed disulfide intermediate. This intermediate is then reduced by the resolving thiol of Trx. (B) In the alternative reaction, the process begins with the same initial step, where NO is transferred to the catalytic thiol, resulting in the release of the reduced peptide. Following this, the bound SNO is reduced by the resolving cysteine, which subsequently releases NO. (See the lines 252-258 in the revise manuscript text))

  1. In Line 300, the sentence should conclude with a full stop (period) to adhere to standard punctuation conventions.

Answer: Yes, the sentence conclude with a full stop.

  1. Please provide a brief description of the function of each enzyme listed in Table 1. This will offer readers a clear understanding of the role these enzymes play in the context of the study.

Answer: Great suggestion, we revised the Table 1 with more detailed data’s such as Functions of enzyme and effect of the S-nitrosation or tyrosine nitration to their activity. (See the line 323 in the revise manuscript text)

Round 2

Reviewer 1 Report

Comments and Suggestions for Authors

I thank the authors for their efforts in responding to my suggestions in the previous revision. However, I still have some comments that could improve the manuscript.

1) L 131. It is not possible to say the presence of nitric oxide synthase because it has not been demonstrated in plants. Authors should specify only the presence of NOS-like activity that has been showed in different papers. Please revise the manuscript to check this.

2) Please, revise all superscript and subscript on the manuscript, examples L143 (NO2), L144 (NO2)

3) L356-357. ‘While both terms Snitrosylation and Snitrosation have been used to describe this PTM, the latter is considered more appropriate’. I consider this statement requires a reference to clarify this controversial in NO field. For instance, authors could check the following paper  

https://doi.org/10.1111/nph.16157 on recommendations for terminology and experimental practice in NO in plants. In any case, authors say in L357 that S nitrosation is more appropriate but in the manuscript they use mainly the expression S-nitrosylation. Please carefully revise this issue.

4) I thanks the authors for their efforts to include a new section in the manuscript but I consider the point about NO effect on seed germination is not necessary since it does not include information on peroxisomes that is the context of this manuscript. My previous comment in revision 1 was about the conclusions section in which authors stated in L436 that ‘investigating the role of S-nitrosylation in ABA function could be a valuable and significant area of study’ and I asked for clarify those aspects to be investigated in this area in the conclusion section. In my opinion, this new point 4.4 could be removed from the new version.

5) L511-512 and L530-531 contain the same information.

6)Please check again the references. In the reference list, number 60, 70 and 77 appears to be the same work and therefore there are some mistakes in the citations in the text. References are very important, especially in a revision manuscript, to facilitate the readers the comprehension of those aspects discussed in the text.

Author Response

We thank to reviewer very much for the constructive comments and suggestions on improving our manuscript. We have carefully considered the comments and suggestions and revised the manuscript to address them. Please find our point-by-point responses below. The paragraphs in blue were our responses. Revised contents in the manuscript were marked in yellow.

Reviewer #1 (Remarks to the Author): I thank the authors for their efforts in responding to my suggestions in the previous revision. However, I still have some comments that could improve the manuscript.

1). L 131. It is not possible to say the presence of nitric oxide synthase because it has not been demonstrated in plants. Authors should specify only the presence of NOS-like activity that has been showed in different papers. Please revise the manuscript to check this.

Answer: Thank you for your special attention to this detail, we specified only the presence of NOS-like activity in the line L131.

2) Please, revise all superscript and subscript on the manuscript, examples L143 (NO2), L144 (NO2)

Answer: Many thanks we agree with your suggestion, we detected some flaws of superscript and subscript in manuscript and we revised them (See the lines L50, L53, L142, L144, L437 and L711).

3) L356-357. ‘While both terms Snitrosylation and Snitrosation have been used to describe this PTM, the latter is considered more appropriate’. I consider this statement requires a reference to clarify this controversial in NO field. For instance, authors could check the following paper  

https://doi.org/10.1111/nph.16157 on recommendations for terminology and experimental practice in NO in plants. In any case, authors say in L357 that S nitrosation is more appropriate but in the manuscript they use mainly the expression S-nitrosylation. Please carefully revise this issue.

Answer: Thank you for your suggestion about to add the reference and revise the lines L356-L357, we agree with your suggestions and we revised these contents (See the lines L353-L357 and L805 in the revise manuscript text).

4) I thanks the authors for their efforts to include a new section in the manuscript but I consider the point about NO effect on seed germination is not necessary since it does not include information on peroxisomes that is the context of this manuscript. My previous comment in revision 1 was about the conclusions section in which authors stated in L436 that ‘investigating the role of S-nitrosylation in ABA function could be a valuable and significant area of study’ and I asked for clarify those aspects to be investigated in this area in the conclusion section. In my opinion, this new point 4.4 could be removed from the new version.

Answer: We appreciate your suggestion, yes we added the new section named “Role of Nitric Oxide (NO) in Seed Germination and Abscisic Acid (ABA) Signaling” in our previous manuscript, in order to make it more coherence and understandable for readers. Furthermore, we added characterizing the interactions and roles of reactive species within peroxisomes, including hydrogen sulfide and nitric oxide, to understand their impact on redox biochemistry and transcriptomic responses to environmental changes. Additionally, investigating nitric oxide’s influence on seed germination, abscisic acid signaling and its interaction with other signaling molecules could provide new insights for enhancing crop resilience and productivity. However, we agree with your opinion so we removed section 4.4 and revised the manuscript.

5) L511-512 and L530-531 contain the same information

Answer: Revised (See the lines L484 in the revise manuscript text).

6)Please check again the references. In the reference list, number 60, 70 and 77 appears to be the same work and therefore there are some mistakes in the citations in the text. References are very important, especially in a revision manuscript, to facilitate the readers the comprehension of those aspects discussed in the text.

Answer: Thank you for your efforts, all the references and citations are checked and revised.
